# Learning Continuous System Dynamics from Irregularly-Sampled Partial Observations

**Zijie Huang**
Department of Computer Science
University of California, Los Angeles
zijiehuang@cs.ucla.edu

**Yizhou Sun**
Department of Computer Science
University of California, Los Angeles
yzsun@cs.ucla.edu

**Wei Wang**
Department of Computer Science
University of California, Los Angeles
weiwang@cs.ucla.edu

## Abstract

Many real-world systems, such as moving planets, can be considered as multi-agent dynamic systems, where objects interact with each other and co-evolve along with the time. Such dynamics is usually difficult to capture, and understanding and predicting the dynamics based on observed trajectories of objects become a critical research problem in many domains. Most existing algorithms, however, assume the observations are regularly sampled and all the objects can be fully observed at each sampling time, which is impractical for many applications. In this paper, we propose to learn system dynamics from irregularly-sampled and partial observations with underlying graph structure for the first time. To tackle the above challenge, we present LG-ODE, a latent ordinary differential equation generative model for modeling multi-agent dynamic system with known graph structure. It can simultaneously learn the embedding of high dimensional trajectories and infer continuous latent system dynamics. Our model employs a novel encoder parameterized by a graph neural network that can infer initial states in an unsupervised way from irregularly-sampled partial observations of structural objects and utilizes neural ODE to infer arbitrarily complex continuous-time latent dynamics. Experiments on motion capture, spring system, and charged particle datasets demonstrate the effectiveness of our approach.

## 1  Introduction

Learning system dynamics is a crucial task in a variety of domains, such as planning and control in robotics [1], predicting future movements of planets in physics [2], etc. Recently, with the rapid development of deep learning techniques, researchers have started building neural-based simulators, aiming to approximate complex system interactions with neural networks [1, 3, 2, 4, 5] which can be learned automatically from data. Existing models, such as Interaction Networks (IN) [3], usually decompose the system into distinct objects and relations and learn to reason about the consequences of their interactions and dynamics based on graph neural network (GNN). However, one major limitation is that they only work for fully observable systems, where the individual trajectory of each object can be accessed at every sampling time. In reality, many applications have to deal with partial observable states, meaning that the observations for different agents are not temporally aligned. For example, when a robot wants to push a set of blocks into a target configuration, only the blocks in the top layer are visible to the camera [1]. More challengingly, the visibility of a specific object might

change over time, meaning that observations can happen at non-uniform intervals for each agent, i.e. irregularly-sampled observations. Such data can be caused by various reasons such as broken sensors, failed data transmissions, or damaged storage [6]. How to learn an accurate multi-agent dynamic system simulator with irregular-sampled partial observations remains a fundamental challenge.

Tang et al. [6] recently have studied a seemingly similar problem which is predicting the missing values for multivariate time series (MTS) as we can view the trajectory of each object as a time series. They assumed there exist some close temporal patterns in many MTS snippets and proposed to jointly model local and global temporal dynamics for MTS forecasting with missing values, where the global dynamics is captured by a memory module. However, it differs from multi-agent dynamic systems in that the model does not assume a continuous interaction among each variable, i.e. the underlying graph structure is not considered. Such interaction plays a very important role in multi-agent dynamic systems which drives the system move forward.

Recently Rubanova et al. [7] has proposed a VAE-based latent ODE model for modeling irregularly-sampled time series, which is a special case of the multi-agent dynamic system where it only handles one object. They assume there exists a latent continuous-time system dynamics and model the state evolution using a neural ordinary differential equation (neural ODE) [8]. The initial state is drawn from an approximated posterior distribution which is parameterized by a neural network and is learned from observations.

Inspired by this, we propose a novel model for learning continuous multi-agent system dynamics under the same framework with GNN as the ODE function to model continuous interaction among objects. However, the main challenge lies in how to approximate the posterior distributions of latent initial states for the whole system, as now the initial states of agents are closely coupled and related to each other. We handle this challenge by firstly aggregating information from observations of neighborhood nodes, obtaining a contextualized representation for each observation, then employ a temporal self-attention mechanism to capture the temporal pattern of the observation sequence for each object. The benefits of joint learning of initial states is twofold: First, it captures the complex interaction among objects. Second, when an object only has few observations, borrowing the information from its neighbors would facilitate the learning of its initial state. We conduct extensive experiments on both simulated and real datasets over interpolation and extrapolation tasks. Experiment results verify the effectiveness of our proposed method.

## 2 Problem Formulation and Preliminaries

Consider a multi-agent dynamic system as a graph $G = \langle O, R \rangle$, where vertices $O = \{o_1, o_2 \cdots o_N\}$ represent a set of $N$ interacting objects, $R = \{\langle i, j \rangle\}$ represents relations. For each object, we have a series of observations $o_i = \{\boldsymbol{o}_i^t\}$ at times $\{t_i^j\}_{j=0}^{T_i}$, where $\boldsymbol{o}_i^t \in \mathbb{R}^D$ denotes the feature vector of object $i$ at time $t$, and $\{t_i^j\}_{j=0}^{T_i}$ can be of variable length and values for each object. Observations are often at discrete spacings with non-uniform intervals and for different objects, observations may not be temporally aligned. We assume there exists a latent generative continuous-time dynamic system, which we aim to uncover. Our goal is to learn latent representations $\boldsymbol{z}_i^t \in \mathbb{R}^d$ for each object at any given time, and utilize it to reconstruct missing observations and forecast trajectories in the future.

### 2.1 Ordinary differential equations (ODE) for multi-agent dynamic system

In continuous multi-agent dynamic system, the dynamic nature of state is described for continuous values of $t$ over a set of dependent variables. The state evolution is governed by a series of first-order ordinary differential equations: $\dot{\boldsymbol{z}}_i^t := \frac{d\boldsymbol{z}_i^t}{dt} = g_i(\boldsymbol{z}_1^t, \boldsymbol{z}_2^t \cdots \boldsymbol{z}_N^t)$ that drive the system states forward in infinitesimal steps over time. Given the latent initial states $\boldsymbol{z}_0^0, \boldsymbol{z}_1^0 \cdots \boldsymbol{z}_N^0 \in \mathbb{R}^d$ for every object, $\boldsymbol{z}_i^t$ is the solution to an ODE initial-value problem (IVP), which can be evaluated at any desired times using a numerical ODE solver such as Runge-Kutta [9]:

$$\boldsymbol{z}_i^T = \boldsymbol{z}_i^0 + \int_{t=0}^T g_i(\boldsymbol{z}_1^t, \boldsymbol{z}_2^t \cdots \boldsymbol{z}_N^t)dt \tag{1}$$

The ODE function $g_i$ specifies the dynamics of latent state and recent works [8, 7, 10] have proposed to parameterize it with a neural network, which can be learned automatically from data. Different

from single-agent dynamic system, $g_i$ should be able to model interaction among objects. Existing works [3, 1, 2, 11] in discrete multi-agent dynamic system employ a shared graph neural network (GNN) as the state transition function. It defines an object function $f_O$ and a relation function $f_R$ to model objects and their relations in a compositional way. By adding residual connection and let the stepsize go to infinitesimal, we can generalize such transition function to the continuous setting as shown in Eqn 2, where $\mathcal{N}_i$ is the set of immediate neighbors of object $o_i$.

$$\dot{\boldsymbol{z}}_i^t := \frac{d\boldsymbol{z}_i^t}{dt} = g_i(\boldsymbol{z}_1^t, \boldsymbol{z}_2^t \cdots \boldsymbol{z}_N^t) = f_O(\sum_{j \in \mathcal{N}_i} f_R([\boldsymbol{z}_i^t, \boldsymbol{z}_j^t])) \tag{2}$$

Given the ODE function, the latent initial state $\boldsymbol{z}_i^0$ for each object determine the whole trajectories.

## 2.2 Latent ODE model for single-agent dynamic system

Continuous single-agent dynamic system is a special case in our setting. Recent work [7] has proposed a latent ODE model following the framework of variational autoencoder [12], where they assume a posterior distribution over the latent initial state $\boldsymbol{z}_0$. The encoder computes the posterior distribution $q\left(\boldsymbol{z}_0 | \{\boldsymbol{o}_i, t_i\}_{i=0}^N\right)$ for the single object with an autoregressive model such as RNN, and sample latent initial state $\boldsymbol{z}_0$ from it. Then the entire trajectory is determined by the initial state $z_0$ and the generative model defined by ODE. Finally the decoder recovers the whole trajectory based on the latent state at each timestamp by sampling from the decoding likelihood $p(\boldsymbol{o}_i | \boldsymbol{z}_i)$.

We model multi-agent dynamic system under the same framework with GNN as the ODE function to model continuous interaction among objects. Since the latent initial states of each object are tightly coupled, we introduce a novel recognition network in the encoder to infer the initial states of all objects simultaneously.

# 3 Method

In this section, we present Latent Graph ODE (LG-ODE) for learning continuous multi-agent system dynamics. Following the structure of VAE, LG-ODE consists of three parts that are trained jointly: 1.) An encoder that infers the latent initial states of all objects simultaneously given their partially-observed trajectories; 2.) a generative model defined by an ODE function that learns the latent dynamics given the sampled initial states. 3.) a decoder that recovers the trajectory based on the decoding likelihood $p(\boldsymbol{o}_i^t | \boldsymbol{z}_i^t)$. The overall framework is depicted in Figure 1. In the following, we describe the three components in detail.

## 3.1 Encoder

Let $\boldsymbol{Z}^t \in \mathbb{R}^{N \times d}$ denotes the latent state matrix of all $N$ objects at time $t$. The encoder returns a factorized distribution of initial states: $q_\phi(\boldsymbol{Z}^0 | o_1, o_2 \cdots o_N) = \prod_{i=1}^N q_\phi(\boldsymbol{z}_i^0 | o_1, o_2 \cdots o_N)$. In multi-agent dynamic system, objects are highly-coupled and related. Instead of encoding temporal pattern for each observation sequence $o_i = \{\boldsymbol{o}_i^t\}_{t=t_i^0}^{t_i^{T_i}}$ independently using an RNN [7], we incorporate structural information by first aggregating information from neighbors' observations, then employ a temporal self-attention mechanism to encode observation sequence for each object. Such process can be decomposed into two steps: 1.) **Dynamic Node Representation Learning**, where we aim to learn an encoding function $f_{\text{update}}$ that outputs structural contextualized representation $\boldsymbol{h}_i^t$ for each observation $\boldsymbol{o}_i^t$. 2.) **Temporal Self-Attention**, where we learn an function $f_{\text{aggre}}$ that aggregates the structural observation representations into a fixed-dimensional sequence representation $\boldsymbol{u}_i$ for each object. $\boldsymbol{u}_i$ is then utilized to approximate the posterior distribution for each latent initial state $\boldsymbol{z}_i^0$.

$$\boldsymbol{h}_i^t = f_{\text{update}}(o_i, \{o_j | \text{if } j \in \mathcal{N}_i\}), \quad \boldsymbol{u}_i = f_{\text{aggre}}(\boldsymbol{h}_i^{t_1}, \boldsymbol{h}_i^{t_2} \cdots \boldsymbol{h}_i^{t_{T_i}}) \tag{3}$$

**Dynamic Node Representation Learning.** One naive way to incorporate structural information is to construct a graph snapshot at each timestamp [13, 14]. However, when system is partially observed, each snapshot may only contain a small portion of objects. For example in Figure 1 (a), 6 out of 7

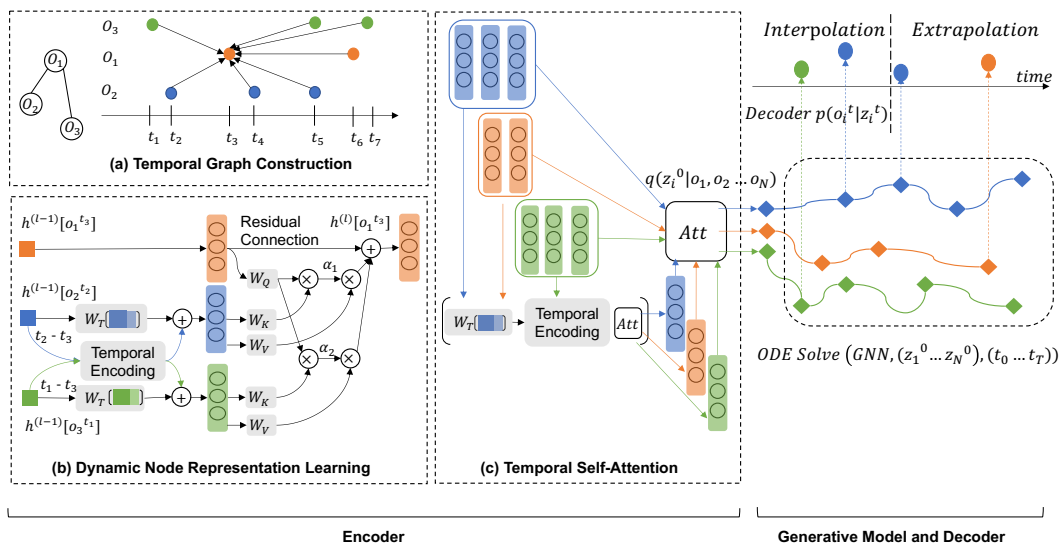

Figure 1: Overall framework.

timestamps only contains one object thus abundant structural information across different snapshots is ignored. We therefore preserve temporal edges and nodes across times to form a temporal graph, where every node is an observation, every edge exists when two objects are connected via a relation $r \in R\{\langle i, j \rangle\}$. Suppose on average every object has $K$ observations, and there are $E$ relations among objects. The constructed temporal graph has $\mathcal{O}(EK^2 + (K-1)KN)$ edges, which grows rapidly with the increase of average observation number $K$. We therefore set a slicing time window that filters out edges when the relative temporal gap is larger than a preset threshold.

To learn a structural representation for each observation, we propose a temporal-aware graph neural network characterized by the information propagation equation in Eqn 4, where $\boldsymbol{h}_t^{l-1}, \boldsymbol{h}_s^{l-1}$ are the representations of target and source node from layer $l-1$ respectively.

$$\boldsymbol{h}_t^l = \boldsymbol{h}_t^{l-1} + \sigma(\sum_{s \in \mathcal{N}_t} (\textbf{Attention}(\boldsymbol{h}_s^{l-1}, \boldsymbol{h}_t^{l-1}) \cdot \textbf{Message}(\boldsymbol{h}_s^{l-1})) \tag{4}$$

To model temporal dependencies among nodes, a simple alternative is to introduce a time-dependent attention score [15] multiply by a linear transformation of the sender node $\boldsymbol{W}_v \boldsymbol{h}_s^{l-1}$. However, the information loss of a sender node w.r.t different temporal gap is linear, as the **Message** of a sender node is time-independent. Recently Transformer [15] has proposed to add positional encoding to the sender node $\boldsymbol{h}_s^{l-1}$ and obtain a time-dependent **Message**. As adding is a linear operator, we hypothesize that taking a nonlinear transformation of the sender node as time-dependent **Message** would be more sufficient to capture the complex nature of information loss caused by temporal gap between nodes. We define the nonlinear transformation w.r.t the temporal gap $\Delta t(s, t)$ as follows:

$$\textbf{Message}(\boldsymbol{h}_s^{l-1}, \Delta t(s,t)) = \boldsymbol{W}_v \hat{\boldsymbol{h}}_s^{l-1}, \quad \hat{\boldsymbol{h}}_s^{l-1} = \sigma(\boldsymbol{W}_t[\boldsymbol{h}_s^{l-1}||\Delta t_{st}]) + \text{TE}(\Delta t_{st})$$

$$\text{TE}(\Delta t)_{2i+1} = cos(\Delta t / 10000^{2i/d}), \quad \text{TE}(\Delta t)_{2i} = sin(\Delta t / 10000^{2i/d}) \tag{5}$$

$$\textbf{Attention}(\boldsymbol{h}_s^{l-1}, \boldsymbol{h}_t^{l-1}, \Delta t(s,t)) = (\boldsymbol{W}_k \hat{\boldsymbol{h}}_s^{l-1})^T (\boldsymbol{W}_q \boldsymbol{h}_t^{l-1}) \cdot \frac{1}{\sqrt{d}}$$

where $||$ is the concatenation operation and $\sigma(\cdot)$ is a non-linear activation function. $d$ is the dimension of node embeddings and $\boldsymbol{W}_t$ is a linear transformation applied to the concatenation of the sender node and temporal gap. We adopt the dot-product form of attention where $\boldsymbol{W}_v, \boldsymbol{W}_k, \boldsymbol{W}_q$ projects input node representations into values, keys and queries. The learned attention coefficient is normalized via softmax across all neighbors. Additionally, to distinguish the sender from observations of the object

itself, and observations from its neighbors, we learn two sets of projection matrices $\boldsymbol{W}_k, \boldsymbol{W}_v$ for each of these two types. Finally, we stack $L$ layers to get the final representation for each observation as $\boldsymbol{h}_i^t = \boldsymbol{h}^L(\boldsymbol{o}_i^t)$. The overall process is depicted in Figure 1 (b).

**Temporal Self-Attention**. To encode temporal pattern for each observation sequence, we design a non-recurrent temporal self-attention layer that aggregates variable-length sequences into fixed-dimensional sequence representations $\boldsymbol{u}_i$, which is then utilized to approximate the posterior distribution for each latent initial state $\boldsymbol{z}_i^0$. Compared with traditional recurrent models such as RNN,LSTM, self-attention mechanism can be better parallelized for speeding up training process and alleviate the vanishing/exploding gradient problem in these models [13, 15]. Note that we have introduced inter-time edges when creating temporal graph, the observation representations $\boldsymbol{h}_i^t$ already preserve temporal dependency across timestamps. To encode the whole sequence, we introduce a global sequence vector $\boldsymbol{a}_i$ to calculate a weighted sum of observations as the sequence representation :

$$\boldsymbol{a}_i = \tanh\left(\left(\frac{1}{N}\sum_t \hat{\boldsymbol{h}}_i^t\right)\boldsymbol{W}_a\right), \quad \boldsymbol{u}_i = \frac{1}{N}\sum_t \sigma(\boldsymbol{a}_i^T \hat{\boldsymbol{h}}_i^t)\hat{\boldsymbol{h}}_i^t \tag{6}$$

where $a_i$ is a simple average of node representations with nonlinear transformation towards the system initial time $t_{\text{start}}$ followed by a linear projection $\boldsymbol{W}_a$. The nonlinear transformation is defined as $\hat{\boldsymbol{h}}_i^t = \sigma(\boldsymbol{W}_t[\boldsymbol{h}_i^t||\Delta t]) + \text{TE}(\Delta t)$ with $\Delta t = (t - t_{\text{start}})$, which is analogous to the **Message** calculation in step 1. Note that if we directly use the observation representation $\boldsymbol{h}_i^t$ from step 1, the sequence representation $\boldsymbol{u}_i$ would be the same when we shift the timestamp for each observation by $T$, as we only utilize the relative temporal gap between observations. By taking the nonlinear transformation, we actually view each observation has an underlying inter-time edge connected to the virtual initial node at system initial time $t_{\text{start}}$. In this way, the sequence representation $\boldsymbol{u}_i$ reflects latent initial state towards a given time $t_{\text{start}}$, and varies when the initial time changes. The process is depicted in Figure 1 (c). Finally, we have the approximated posterior distribution as in Eqn7 where $f$ is a neural network translating the sequential representation into the mean and variance of $\boldsymbol{z}_i^0$.

$$q_\phi(\boldsymbol{z}_i^0|o_1, o_2 \cdots o_N) = \mathcal{N}\left(\boldsymbol{\mu}_{z_i^0}, \boldsymbol{\sigma}_{z_i^0}\right), \quad \text{where } \boldsymbol{\mu}_{z_i^0}, \ \boldsymbol{\sigma}_{z_i^0} = f(\boldsymbol{u}_i) \tag{7}$$

### 3.2 Generative model and decoder

We consider a generative model defined by an ODE whose latent initial state $\boldsymbol{z}_i^0$ is sampled from the approximated posterior distribution $q_\phi(\boldsymbol{z}_i^0|o_1, o_2 \cdots o_N)$ from the encoder. We employ a graph neural network (GNN) in Eqn 2 as the ODE function $g_i$ to model the continuous interaction of objects. A decoder is then utilized to recover trajectory from the decoding probability $p(\boldsymbol{o}_i^t|\boldsymbol{z}_i^t)$, characterized by a neural network.

$$\begin{gathered}
\boldsymbol{z}_i^0 \sim p(\boldsymbol{z}_i^0) \approx q_\phi(\boldsymbol{z}_i^0|o_1, o_2 \cdots o_N) \\
\boldsymbol{z}_i^0, \boldsymbol{z}_i^1 \cdots \boldsymbol{z}_i^T = \text{ODESolve}(g_i, [\boldsymbol{z}_1^0, \boldsymbol{z}_2^0 \cdots \boldsymbol{z}_N^0], (t_0, t_1 \cdots t_T)) \\
\boldsymbol{o}_i^t \sim p(\boldsymbol{o}_i^t|\boldsymbol{z}_i^t)
\end{gathered} \tag{8}$$

### 3.3 Training

We jointly train the encoder, decoder and generative model by maximizing the evidence lower bound (ELBO) as shown below. As observations for each object are not temporally aligned in a minibatch, we take the union of these timestamps and output the solution of the ODE at them.

$$\begin{aligned}
&ELBO(\theta, \phi) \\
&= \mathbb{E}_{\boldsymbol{Z}^0 \sim q_\phi(\boldsymbol{Z}^0|o_1, \cdots o_N)}[\log p_\theta(o_1, \ldots, o_N)] - \text{KL}[q_\phi(\boldsymbol{Z}^0|o_1, \cdots, o_N)\|p(\boldsymbol{Z}^0)] \\
&= \mathbb{E}_{\boldsymbol{Z}^0 \sim \prod_{i=1}^N q_\phi(\boldsymbol{z}_i^0|o_1, \cdots, o_N)}[\log p_\theta(o_1, \cdots, o_N)] - \text{KL}[\prod_{i=1}^N q_\phi(\boldsymbol{z}_i^0|o_1, \cdots, o_N)\|p(\boldsymbol{Z}^0)]
\end{aligned} \tag{9}$$

# 4 Experiments

## 4.1 Datasets

We illustrate the performance of our model on three different datasets: particles connected by springs, charged particles ( Kipf et al. [2]) and motion capture data ( CMU [16]). The first two are simulated datasets, where each sample contains 5 interacting particles in a 2D box with no external forces (but possible collisions with the box). The trajectories are simulated by solving two types of motion PDE for spring system and charged system respectively [2] with the same number of forward steps 6000 and then subsampling each 100 steps. To generate irregularly-sampled partial observations, for each particle we sample the number of observations $n$ from $\mathcal{U}(40, 52)$ and draw the $n$ observations uniformly from the PDE steps to get the training trajectories. To evaluate extrapolation task, we additionally sample 40 observations following the same procedure from PDE steps $[6000, 12000]$ for testing. The above sampling procedure is conducted independently for each object. We generate 20k training samples and 5k testing samples for these two datasets respectively. For motion capture data, we select the walking sequences of subject 35. Every sample is in the form of 31 trajectories, each tracking a single joint. Similar as simulated datasets, for each joint we sample the number of observations $n$ from $\mathcal{U}(30, 42)$ and draw the $n$ observations uniformly from first 50 frames for training trajectories. For testing, we additionally sampled 40 observations from frames $[51, 99]$. We split the different walking trials into non-overlapping training (15 trials) and test sets (7 trials).

We conduct experiment on both interpolation and extrapolation tasks as proposed in Rubanova et al. [7]. For all experiments, we report the mean squared error (MSE) on the test set. For all datasets, we rescale the time range to be in $[0, 1]$. Our implementation is available online[1]. More details can be found in the supplementary materials.

## 4.2 Baselines and Model Variants

**Baselines.** To the best of our knowledge, existing works on modeling multi-agent dynamic system with underlying graph structure cannot handle irregularly-sampled partial observations, in which these models require full observation at timestamp $t$ in order to make prediction at timestamp $t + 1$ [2, 3]. Therefore, we firstly compare our model with different encoder structures to infer the initial states. Specifically, we consider Latent-ODE (Rubanova et al. [7]) which has shown to be successful for encoding single irregularly-sampled time series without considering graph interaction among agents. Edge-GNN (Gong and Cheng [17]) incorporates temporal information by viewing time gap as an edge attribute. Weight-Decay considers a simple exponential decay function for time gap as similar in Cao et al. [18], which models $\boldsymbol{h}(t + \Delta t) = exp\{-\tau \Delta t\} \cdot \boldsymbol{h}(t)$ with a learnable decay parameter $\tau$. The sequence representation of Edge-GNN and Weight-Decay is the weighted sum of observations within a sequence. We additionally compare LG-ODE against an RNN-based MTS model for handling irregularly-sampled missing values [19] where the graph structure is not considered. It jointly imputes missing values for all agents by simple concatenation of their feature vectors. We compare it in the Interpolation Task which is to imputes missing values within the observed sequences. After imputation, we employ NRI [2] which is a multi-agent dynamic system model with regular observations and graph input to predict future sequences. We refer to this task as Extrapolation Task. In what follows, we refer to the combination of these two models as RNN-NRI.

**Model Variants.** Our proposed encoder contains two modules: dynamic node representation network followed by a temporal self-attention. To further analyze the components within each module, we conduct an ablation study by considering five model variants. Firstly, module one contains two core components: attention mechanism and learnable positional encoding within GNN for capturing temporal and spatial dependency among nodes. We therefore remove them separately and get LG-ODE-no att, LG-ODE-no PE respectively. We additionally compare our learnable positional encoding with manually-designed positional encoding [15] denoted as LG-ODE-fixed PE. Secondly, we apply various sequence representation methods to test the efficiency of module two: LG-ODE-first takes the first observation in a sequence as sequence representation and LG-ODE-mean uses the mean pooling of all observations as sequence representation.

Table 1: Mean Squared Error(MSE) $\times 10^{-2}$ on Interpolation task.

| | Springs | | | Charged | | | Motion | | |
|---|---|---|---|---|---|---|---|---|---|
| Observed ratio | 40% | 60% | 80% | 40% | 60% | 80% | 40% | 60% | 80% |
| Latent-ODE | 0.5454 | 0.5036 | 0.4290 | 1.1799 | 1.1198 | 0.8332 | 0.7709 | 0.4826 | 0.3603 |
| Weight-Decay | 1.1634 | 1.1377 | 1.6217 | 2.8419 | 2.2547 | 1.5390 | 1.9007 | 2.0023 | 1.6894 |
| Edge-GNN | 1.3370 | 1.2786 | 0.8188 | 1.5795 | 1.5618 | 1.1420 | 2.7670 | 2.6582 | 1.8485 |
| NRI + RNN | 0.5225 | 0.4049 | 0.3548 | 1.3913 | 1.1659 | 1.0344 | 0.5845 | 0.5395 | 0.5204 |
| LG-ODE | **0.3350** | **0.3170** | **0.2641** | **0.9234** | **0.8277** | **0.8046** | 0.4515 | **0.2870** | **0.3414** |
| LG-ODE-first | 1.3017 | 1.1918 | 1.0796 | 2.5105 | 2.6714 | 2.3208 | 1.4904 | 1.3702 | 1.2107 |
| LG-ODE-mean | 0.3896 | 0.3901 | 0.3268 | 1.1246 | 1.0050 | 0.9133 | 0.6415 | 0.5834 | 0.5549 |
| LG-ODE-no att | 0.5145 | 0.4198 | 0.4510 | 0.9372 | 0.9503 | 0.9752 | 0.6991 | 0.6998 | 0.7452 |
| LG-ODE-no PE | 0.4431 | 0.4278 | 0.3879 | 1.0450 | 1.0350 | 0.9621 | 0.4677 | 0.4808 | 0.4799 |
| LG-ODE-fixed PE | 0.4285 | 0.4445 | 0.4083 | 0.9838 | 0.9775 | 0.9524 | **0.4215** | 0.4371 | 0.4313 |

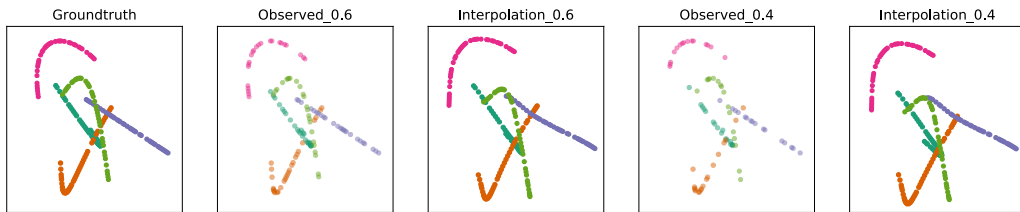

Figure 2: Visualization of interpolation results for spring system.

## 4.3 Results on Interpolation Task

**Set up.** In this task, we condition on a subset of observations $(40\%, 60\%, 80\%)$ from time $(t_0, t_N)$ and aim to reconstruct the full trajectories in the same time range. We subsample timepoints for each object independently.

Table 1 shows the interpolation MSE across different datasets and methods. Latent-ODE performs well on encoding single timeseries but fails to consider the interaction among objects, resulting in its poor performance in the multi-agent dynamic system setting. Weight-Decay and Edge-GNN utilize fixed linear transformation of sender node to model information loss across timestamps, which is not sufficient to capture the complex temporal dependency. RNN-NRI though handles the irregular temporal information by a specially designed decay function, it conducts imputation without considering the graph interaction among objects and thus obtaining a poor performance. By comparing model variants for temporal self-attention module, we notice that taking the first observation as sequence representation produces high reconstruction error, which is expected as the first observable time for each sequence may not be the same so the inferred latent initial states are not aligned. Averaging over observations assumes equal contribution for each observation and ignores the temporal dependency, resulting in its poor performance. For module one, experiment results on model variants suggest that distinguishing the importance of nodes w.r.t time and incorporating temporal information via learnable positional encoding would benefit model performance. Notably, the performance gap between LG-ODE and other methods increases when the observation percentage gets smaller, which indicates the effectiveness of LG-ODE on sparse data. When observation percentage increases, the reconstruction loss of all models tends to be smaller, which is expected. Figure 2 visualizes the interpolation results of our model under different observation percentage for the spring system. Figure 3 visualizes the interpolation results for motion capture data with 60% observation percentage.

## 4.4 Results on Extrapolation Task

**Set up.** In this task we split the time into two parts: $(t_0, t_{N_1})$ and $(t_{N_1}, t_N)$. We condition on the first half of observations and reconstruct the second half. For training, we condition on observations from $(t_1, t_2)$ and reconstruct the trajectories in $(t_2, t_3)$. For testing, we condition on the observations from $(t_1, t_3)$ but tries to reconstruct future trajectories within $(t_3, t_4)$. Similar to interpolation task, we experiment on conditioning only on a subset of observations in the first half and run the encoder

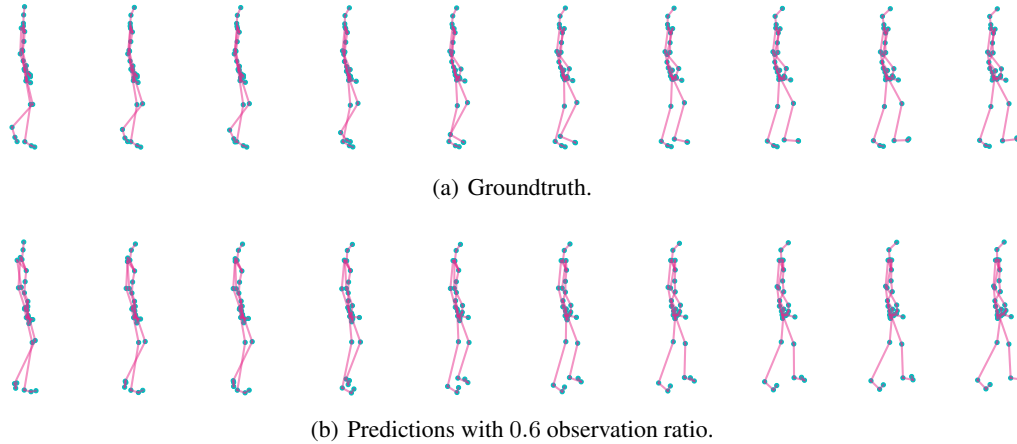

(a) Groundtruth.

(b) Predictions with 0.6 observation ratio.

Figure 3: Visualization of interpolation results for walking motion data.

on the subset to estimate the latent initial states. We evaluate model's performance on reconstructing the full trajectories in the second half.

Table 2: Mean Squared Error(MSE) $\times 10^{-2}$ on Extrapolation task.

| Extrapolation | Springs | | | Charged | | | Motion | | |
|---|---|---|---|---|---|---|---|---|---|
| Observed ratio | 40% | 60% | 80% | 40% | 60% | 80% | 40% | 60% | 80% |
| Latent-ODE | 6.6923 | 4.2478 | 4.3192 | 13.5852 | 12.7874 | 20.5501 | 2.4186 | 2.9061 | 2.6590 |
| Weight-Decay | 6.1559 | 5.7416 | 5.3712 | 9.4764 | 9.1008 | 9.0886 | 16.8031 | 13.6696 | 13.6796 |
| Edge-GNN | 6.0417 | 4.9220 | 3.2281 | 9.2124 | 9.1410 | 8.8341 | 13.2991 | 13.9676 | 9.8669 |
| NRI + RNN | 2.6638 | 2.4003 | 2.5550 | 7.1776 | 6.9882 | 6.6736 | 3.5380 | 3.0119 | 2.6006 |
| LG-ODE | **1.7839** | 1.8084 | **1.7139** | 6.5320 | **6.4338** | **6.2448** | **1.2843** | **1.2435** | 1.2010 |
| LG-ODE-first | 6.5742 | 6.3243 | 5.7788 | 9.3782 | 9.2107 | 8.4765 | 3.8864 | 3.2849 | 3.0001 |
| LG-ODE-mean | 2.2499 | 2.1165 | 2.2516 | 9.1355 | 8.7820 | 8.4422 | 1.3169 | 1.3008 | 1.2534 |
| LG-ODE-no att | 2.3847 | 2.1216 | 1.9634 | 7.2958 | 7.3609 | 6.7026 | 3.4510 | 3.2178 | 3.9917 |
| LG-ODE-no PE | 1.7943 | 1.8172 | 1.7332 | 6.9961 | 6.7208 | 6.5852 | 1.5054 | 1.2997 | 1.2029 |
| LG-ODE-fixed PE | 1.7905 | **1.7634** | 1.7545 | **6.4520** | 6.4706 | 6.3543 | 1.4624 | 1.2517 | **1.1992** |

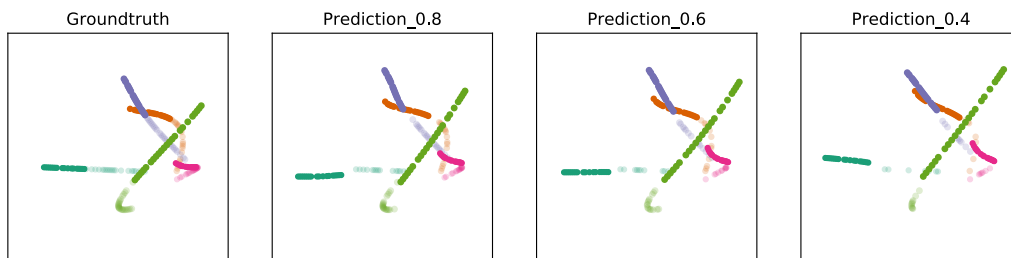

Figure 4: Visualization of extrapolation results for spring system. Semi-transparent paths denote observations from first-half of time, from which the latent initial states are estimated. Solid paths denote model predictions.

Table 2 shows the MSE on extrapolation task. The average MSE in extrapolation task is greater than interpolation task, which is expected as predicting the future is a more challenging task. Similar as in interpolation task, when observation percentage increases, the prediction error of all models tends to become smaller. LG-ODE achieves better results across different datasets and settings, which verifies the effectiveness of our design to capture structural dependency among objects, and temporal dependency within observation sequence. Specifically, RNN-NRI is a two-step model that first imputes each time series into regular-sampled one to make it a valid input for NRI, and then predict trajectories with the graph structure. LG-ODE instead is an end-to-end framework. The

prediction error for RNN-NRI is large and one possible reason is that we use estimated imputation values for missing data which would add noise to NRI. We also notice that the performance drop due to the sparsity of observations is small in LG-ODE compared with other baselines, which shows our model is more powerful especially when data is sparse. We illustrate the predicted trajectories of spring system under different observation percentage as shown in Figure 4.

## 5 Related Work

**Neural Physical Simulator**. Existing works have developed various neural-based physical simulators that learn the system dynamics from data [1, 3]. In particular, Kipf et al. [2] and Battaglia et al. [3] have explored learning a simulator by approximating pair-wise object interactions with graph neural networks. These approaches restrict themselves to learn a fixed-step state transition function that takes the system state at time $t$ as input to predict the state at time $t + 1$. However, they can not be applied to the scenarios where system observations are irregularly sampled. Our model handles such issue by combining a neural ODE [8] to model continuous system dynamics and a temporal-aware graph neural network followed by a temporal self-attention module to estimate system initial states. Another issue lies in that they need to observe the full states of a system; but in reality, system states are often partially observed where number and set of observable objects vary over time. A recent work [1] tackled this issue where system is partially observed but observations are regularly sampled by learning the dynamics over a latent global representation for the system, which is for example an average over the sets of object states. However it cannot directly learn the dynamic state for each object. In our work, we design a dynamic model that explicitly operates on the latent dynamic representations over each object. This allows us to define object-centric dynamics, which can better capture system dynamics compared to the coarse global system representation.

**Dynamic Graph Representation**. Our model takes the form of variational auto-encoder. The encoder which is used to infer latent initial states is closely related to dynamic graph representation. Most existing methods [13, 20, 14] learn the dynamic representations of nodes by splitting the input graph into snapshot sequence based on timestamps [21]. Each snapshot is passed through a graph neural network to capture structural dependency among neighbors and then a recurrent network is utilized to capture temporal dependency by summarizing historical snapshots. However recurrent methods scale poorly with the increase in number of time-steps [13]. Moreover, when system states are partially observed, each timestamp may only contain a small portion of objects and abundant structural information across different snapshots is ignored. A recent work [21] proposed to maintain all the edges happening in different times as a whole and introduced relative temporal encoding strategy (RTE) to model structural temporal dependencies with any duration length. RTE utilizes a linear transformation of the sender node with regard to a given timestamp based on positional encoding [15]. In our work, we explored a more complex nonlinear transformation to capture complex temporal dependency in dynamic physical system.

## 6 Discussion and Conclusion

In this paper, we propose LG-ODE for learning continuous multi-agent system dynamics from irregularly-sampled partial observations. We model system dynamics through a neural ordinary differential equation and draw the latent initial states for each object simultaneously through a novel encoder that is able to capture the interaction among objects. The joint learning of initial states not only captures interaction among objects but can benefit the learning when an object only has few observations. We achieve state-of-the-art performance in both interpolating missing values and extrapolating future trajectories. An limitation of current model is that we assume the underlying interaction graph is fixed over time. In the future, we plan to learn the system dynamics when the underlying interaction graph is evolving.

## Broader Impact

Learning system dynamics is an important task in various of fields such as biology, physics, robotics, etc. Existing models only work for fully observable systems, requiring the observations of all object at each sample timestamp. However in reality, data is usually incomplete due to various reasons such as broken sensors. More challengingly, observations can happen at non-uniform intervals. Our model is able to learn system dynamics from such irregularly-sampled partial observations, and can be applied to various applications such as planning and control in robotics especially when data is incomplete.

## Acknowledgement

This work is partially supported by NSF III-1705169, NSF CAREER Award 1741634, NSF 1937599, DARPA HR00112090027, Okawa Foundation Grant,Amazon Research, NSF DBI-1565137, DGE-1829071, IIS-2031187, NIH R35-HL135772, NEC Research Gift, and Verizon Media Faculty Research and Engagement Program.

## Footnotes

[1]https://github.com/ZijieH/LG-ODE.git

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
