[Supplementary Material]

# Supplementary for Learning Continuous System Dynamics from Irregularly-Sampled Partial Observations

**Zijie Huang**
Department of Computer Science
University of California, Los Angeles
zijiehuang@cs.ucla.edu

**Yizhou Sun**
Department of Computer Science
University of California, Los Angeles
yzsun@cs.ucla.edu

**Wei Wang**
Department of Computer Science
University of California, Los Angeles
weiwang@cs.ucla.edu

## 1   Experiment Setup

We evaluate the performance of LG-ODE across different datasets on two tasks: Interpolation and Extrapolation. For each task, there exists a special time $t_{start}$ where we put a prior $p(z_i^0) \approx q_\phi(z_i^0|o_1, o_2 \cdots o_N)$ on for each object. We now present the normalization across times and the selection of $t_{start}$ for each task respectively.

### 1.1   Interpolation

In this task, we condition on a subset of observations $(40\%, 60\%, 80\%)$ from time $(t_0, t_N)$ and aim to reconstruct the full trajectories in the same time range. In single-agent dynamic system [1], $t_{start}$ is the time point of the first observation in each sequence. However in multi-agent dynamic system, the first observable time point for each object may differ. We therefore define a system starting time $t_{start} = 0$ as the initial time point, and normalize all the observation times within $[0, 1]$. In this way, we assume the continuous interaction among objects starts at $t_{start} = 0$ and therefore solve the ODE forward in time range $[0, 1]$.

### 1.2   Extrapolation

In this task we split the time into two parts: $(t_0, t_{N_1})$ and $(t_{N_1}, t_N)$. We condition on the first half of observations and reconstruct the second half. In other words, we solve the ODE forward in time range $[t_{N_1}, t_N]$ with $t_{N_1}$ as $t_{start}$. For training, we condition on observations from $(t_1, t_2)$ and reconstruct the trajectories in $(t_2, t_3)$. For testing, we condition on the observations from $(t_1, t_3)$ but tries to reconstruct future trajectories within $(t_3, t_4)$.

For training, as we sample each whole sequence from $[t_1, t_3]$, we manually separate each sequence into two halves by choosing $t_{start} = t_{N_1} = \frac{t_1 + t_3}{2}$ as system starting time. Observations with time less than $t_{start}$ are sent into encoder to estimate the latent initial states, observations with time equal are greater than $t_{start}$ are viewed as ground truth for reconstructing trajectories in the second half. For testing, as we additionally sampled 40 observations in a wider time range $[t_3, t_4]$, we set $t_{start} = t_{N_1} = t_3$ and try to reconstruct the full trajectories on them. Observations with time less than $t_3$ are used to infer latent initial state. Similar to interpolation task, we experiment on conditioning only on a subset of observations $(40\%, 60\%, 80\%)$ in the first half and run the encoder on the subset to estimate the latent initial states. We normalize all the observation times within $[0, 1]$.

Figure 1: Examples of trajectories with 5 objects.

# 2 Data Generation and Preprocessing

## 2.1 Simulated Datasets

We generate two simulated datasets: particles connected by springs and charged particles based on Kipf et al. [2]. Each sample system contains 5 interacting objects. For the spring system, objects do or do not interact with equal probability and interact via forces given by Hooke's law. For the charged particles, they attract or repel with equal probability. Trajectories are simulated by leapfrog integration with a fixed step size and we subsample each 100 steps to get trajectories. For training, we use total 6000 forward steps. To generate irregularly-sampled partial observations, for each object we sample the number of observations $n$ from $\mathcal{U}(40, 52)$ and draw the $n$ observations uniformly. For testing, besides generating observations from steps $[1, 6000]$, we additionally sample 40 observations following the same procedure from PDE steps $[6000, 12000]$, to evaluate extrapolation task. The above sampling procedure is conducted independently for each object. We generate 20k training samples and 5k testing samples for two datasets respectively. We normalize all features (position/velocity) to maximum absolute value of 1 across training and testing datasets. Example trajectories can be seen in Figure 1.

## 2.2 CMU walking capture dataset

For motion capture data, we select the walking sequences of subject 35 from CMU [3]. Every sample is in the form of 31 trajectories, each tracking a single joint. Similar as simulated datasets, for each joint we sample the number of observations $n$ from $\mathcal{U}(30, 42)$ and draw the $n$ observations uniformly from first 50 frames for training trajectories. For testing, we additionally sampled 40 observations from frames $[51, 99]$. We split different walking sequences into training (15 trials) and test sets (7 trials). For each walking sequence, we further split it into several non-overlapping small sequences with maximum length 50 for training, and maximum length 100 for testing. In this way, we generate total 120 training samples and 27 testing samples. We normalize all features (position/velocity) to maximum absolute value of 1 across training and testing datasets.

# 3 Model Architecture and Hyperparameters

## 3.1 ODE function

The ODE function $g_i$ specifies the dynamics of latent state and we employ a graph neural network (GNN) in [2] to model interaction among objects. It defines an object function $f_O$ and a relation function $f_R$ to model objects and their relations in a compositional way. Following the default setting

in [2], given all the edges in an interaction graph, we also consider latent interactions among objects where no edge is observed. More precisely, we have:

$$\dot{z}_i^t := \frac{dz_i^t}{dt} = g_i(z_1^t, z_2^t \cdots z_N^t) = f_O\big( \sum_{j \in \mathcal{N}_i} f_R([z_i^t, z_j^t]) + \sum_{j \notin \mathcal{N}_i} f_R([z_i^t, z_j^t]) \big)$$

$$f_R([z_i^t, z_j^t]) = e_{ij} = \begin{cases} MLP_r^0([z_i^t || z_j^t]) & for\ j \in \mathcal{N}_i \\ MLP_r^1([z_i^t || z_j^t]) & for\ j \notin \mathcal{N}_i \end{cases} \tag{1}$$

$$f_O\big( \sum_{j \in \mathcal{N}_i} f_R([z_i^t, z_j^t]) + \sum_{j \notin \mathcal{N}_i} f_R([z_i^t, z_j^t]) \big) = MLP_o\big( \sum_{j \in \mathcal{N}_i} e_{ij} + \sum_{j \notin \mathcal{N}_i} e_{ij} \big)$$

where $\mathcal{N}_i$ is the set of immediate neighbors of object $o_i$, $||$ is the concatenation operations, $f_O, f_R$ are two feed-forward multi-layer perception neural nets with Relu as activation function. For all datasets, we use a one-layer GNN with 128-dim hidden node embeddings. To stablize training process, we use the idea of [4] and add auxiliary 64-dimensions to the learned initial hidden representation $z_i^t$ from the encoder.

## 3.2   ODE solver

We use the fourth order Runge-Kutta method from torchdiffeq python package [5] as the ODE solver, for solving the ODE systems on a time grid that is five times denser than the observed time points. We also utilize the Adjoint method described in [5] to reduce the memory use.

## 3.3   Recognition Network

We next introduce model details in the encoder, which aims to infer latent initial states for all objects simultaneously.

**Temporal Graph Edge Sampling**. We preserve all temporal edges and nodes across times to form a temporal graph, where every node is an observation, every edge exists when two object are connected via a relation. Suppose on average every object has $K$ observations, and there are $E$ relations among objects. The constructed temporal graph has $O(EK^2 + (K-1)KN)$ edges, which grows rapidly with the increase of average observation number $K$. We therefore set a slicing time window that filters out edges when the relative temporal gap is larger than a preset threshold. As $K$ is related to observation percentage, we set the threshold accordingly as :

$$\frac{Max\_Time\_Length - Min\_Time\_Length \times \%observed}{Max\_Time\_Length} \tag{2}$$

where $Max\_Time\_Length$ and $Min\_Time\_Length$ denotes the maximum observation sequence length and minimum observation sequence length among objects. When observation percentage gets larger, we tend to filter more edges by setting a relatively small threshold.

**Dynamic Node Representation Learning and Temporal Self-attention**. We set the observation representation dimension from the GNN as 64 and keep 2 layers across all datasets. We use Relu as the activation function. For temporal self-attention module, we set the output dimension as 128.

We implement our model in pytorch. Encoder, generative model and the decoder parameters are jointly optimized with Adam optimizer [6] with learning rate 0.0005. Batch size for simulated datsests are set as 256, and is 32 for motion dataset. Code is available online[1].

## Footnotes

[1]https://github.com/ZijieH/LG-ODE.git

[2] Thomas Kipf, Ethan Fetaya, Kuan-Chieh Wang, Max Welling, and Richard Zemel. Neural relational inference for interacting systems. *arXiv preprint arXiv:1802.04687*, 2018.

[3] CMU. Carnegie-mellon motion capture database. 2003. URL `http://mocap.cs.cmu.edu`.

[4] Emilien Dupont, Arnaud Doucet, and Yee Whye Teh. Augmented neural odes. In *Advances in Neural Information Processing Systems 32*, pages 3140–3150. 2019.

[5] Ricky T. Q. Chen, Yulia Rubanova, Jesse Bettencourt, and David K Duvenaud. Neural ordinary differential equations. In *Advances in Neural Information Processing Systems 31*, pages 6571–6583. 2018.

[6] Diederik P. Kingma and Jimmy Ba. Adam: A method for stochastic optimization. *ICLR*, 2015.

## References

[1] Yulia Rubanova, Ricky T. Q. Chen, and David K Duvenaud. Latent ordinary differential equations for irregularly-sampled time series. In *Advances in Neural Information Processing Systems 32*, pages 5320–5330. 2019.