[Reviews · NeurIPS 2020]

Review 1

Summary and Contributions: This paper proposes a novel approach to learning system dynamics in the general case where the time between observations is non-uniform and where the observations from different objects are not temporally aligned. The main contribution is the design of an architecture capable of handling this case. To this end, the paper proposes an encoding scheme that infers latent initial states of all objects simultaneously. Starting from these latent initial states an ODE function models the latent dynamics, which are subsequently decoded into the observed space. The entire architecture can be learned end-to-end using variational optimization.

Strengths: - The paper is timely and addresses an important topic that is of high interest to the machine learning community. I believe this is the first time that a paper models irregular time and partial observations in the context of latent multi-agent dynamics. At the same time, this is a common case in many real-world scenarios. - The high-level design of the presented architecture and the interactions between graph encoding, ODE dynamics, and variational optimization are clean and elegant. - The presented approach to dynamic node representation learning (section 3.1) is appealing and a major contribution of the paper. The low-level details feel at times a bit overengineered (see below). - The experimental evaluation of the presented approach poses the problem that no real baselines exist. The authors decided to focus on alternative encoding schemes, which is reasonable, and at least in these cases the results show a clear edge of the proposed model in terms of performance.

Weaknesses: - While ultimately a great contribution, the section on dynamic node representation learning is quite complex; the same net effect could potentially be achieved with a simpler design. An ablation study isolating some of the presented components would convince the reader of their necessity. - The experimental evaluation is limited to simple toy datasets and motion capture sequences from a single subject. I would have liked to see a more comprehensive analysis of additional mocap sequences or other real-world problems. Furthermore, results highlighting the interaction aspect of the model are missing. - Given that this is a paper about temporal dynamics, it is surprising that no video sequences are included in the supplemental material. For instance, it would be very interesting to see the walking sequence shown in Fig. 3 in motion. - The role of g_i (or f_O/f_R, to be more precise) remains vague. It is introduced in Eq.(2) and appears again in the generative model (Eq.(8)), but no details regarding its design or structure are provided. - The limitations of the model are not adequately discussed. For instance, it is not clear which of the common inference tasks beyond interpolation/extrapolation are feasible with the proposed model.

Correctness: The presented framework appears to be sound. The few plots/equations that made me hesitate are likely just typos or have a simple explanation: (1) The variables indicating the latent initial states (three lines before Eq.(1)) are wrong. (2) The second argument of “Attention(.)” in Eq.(5) should have “t” as a subscript. (3) In Fig. 2 (interpolation 0.4), why do the interpolated sequences (e.g., light green) not agree with the observed points they are conditioned on?

Clarity: The paper is well written and for the most part easy to follow. The structure is clear and the goals and objectives are well motivated. The notation and terminology are consistent. The overview figure (Fig. 1) is insightful and guides the reader through section 3. The section on dynamic node representation learning is initially a bit confusing, for instance, the variable h has a subscript that refers to an object or source/target, depending on the context, and a superscript that refers to time or layer, again depending on the context. Some of the equations, e.g., TE in Eq.(5) or u_i in Eq.(6) are not discussed in the text.

Relation to Prior Work: The discussion of related work is adequate but close to the lower bar; it focuses mainly on the object interaction portion of the paper. I would have liked to see a discussion of the topic in the broader context of latent time-series dynamics, which is a line of research with a very long history. I encourage the authors to add some historical context.

Reproducibility: Yes

Additional Feedback: Post-rebuttal comment: After reading the rebuttal and the other reviews, I will keep my original rating (7). It's a well-written paper with contributions that would be of interest to the community.


Review 2

Summary and Contributions: This paper introduces a new method for learning continuous system dynamics when observations are irregularly sampled and observed. The method applies to collective multi-agent dynamics, for instance, several bouncing balls in a box. The proposed learning architecture combines neural ODEs and graph neural networks to model continuous forward dynamics, and crucially, introduces a new network module to infer the initial state of all objects. Because the initial state governs the evolution of the ODE, estimating it well is critical. The proposed method contextualizes each observation with nearby objects in time and uses a temporal self attention component to decide on an initial state. Experimental results show that the proposed method outperforms simpler baselines that do not consider multiagent interactions, on several sythetic datasets and one motion capture dataset.

Strengths: This study treats an important problem and proposes an interesting architecture to solve a key piece of it--determining the initial state of a dynamical system from partial, irregular observations. This is a novel contribution that would be relevant to the NeurIPS community. It seems likely that subsequent work would build on this starting point. The experimental results are conducted to a reasonably high standard and show the benefits of the proposed method over other baseline algorithms, both for interpolating between the irregular observations and for extrapolating forward in time well beyond them. The synthetic examples are useful in showing the basic advantages of the approach, and the motion capture dataset shows that the approach might scale to more realistic data.

Weaknesses: The evaluations in the paper focus on relatively simple settings compared to other applications of graph neural networks. The paper would be stronger with applications to richer domains, but in my view, the evaluations given are enough to show the promise of the method. I found the claim in the abstract that "we propose to learn system dynamics from irregularly-sampled and partially observed system for the first time" to be too strong. Other methods have addressed irregular samples and partial observation, using EM-style algorithms. (For instance, Nguyen et al., EM-like Learning Chaotic Dynamics from Noisy and Partial Observations. Arxiv (2019), or Clinical time series prediction: towards a hierarchical dynamical system framework. Artif Intell Med (2015).) I would not be surprised to learn of other earlier work in this direction. The proposed method is novel in the specific algorithm it offers, not the problem it tackles.

Correctness: The methodology appears correct and sound.

Clarity: The paper is clearly written.

Relation to Prior Work: Prior work is well cited, although as noted above, I think this overall problem has been treated by some prior work using very different methods.

Reproducibility: Yes

Additional Feedback: The broader impact statement focuses on positive uses of the methodology, but could be balanced with potential negative uses of the method.


Review 3

Summary and Contributions: Authors propose a neural network parameterized continuous-time dynamical model that can learn ODE models from irregularly and partially sampled data. Briefly, the proposed method builds on/combines VAE and neural ODE models and simultaneously learns an encoding/decoding model for high-dimensional data and models the continuous-time dynamics in a latent space. While similar methods have been proposed in the literature recently, this method extends the method to allow learning from part irregularly and partially sampled data. The problem that the authors tackle is important as many real world problems are naturally modelled using ODEs/SDEs/PDEs but often the driving function is unknown and must be learned from observed data. The method is formulated as a system of continuous-time interacting agents, but the method falls under the general umbrella of continuous-time dynamical systems considered e.g. in neural ODE and several other papers; here the method can assume possibly useful additional information about interactions between agents/variables.

Strengths: As authors note, this is probably the first and only method proposed so far that can learn NN parameterized ODEs from irregularly and partially sampled data, which is the main strength of the manuscript. Authors demonstrate that under those assumptions, they obtain better results than competing methods (which strictly speaking there are none as other methods have not assumed irregularly-sampled and partial data).

Weaknesses: I consider that the main weakness of the manuscript is a proper ablation study that would provide a comprehensive characterization of the capabilities of the proposed ODE and encoder models. By ablation study I mean a comparison with previous methods (such as neural ODEs and other methods cited in the manuscript) in a setting where the other methods can be reliably applied, including fully-observed data as well as regularly sampled data. While I appreciate that the method can work with irregularly/partially observed data, it is difficult to assess that how useful and accurate e.g. the proposed encoder is when compared to previous methods, when e.g. only a few data points are missing which could be reliably imputed using any state-of-the-art missing value imputation/interpolation method before applying current continuous-time modeling methods to regularly-sampled/full data. Authors should elaborate the benefit/drawback of their modeling assumption in Eq. (2) which shows that the latent state z_i of each "agent" is driven by the same driving function f_O(). In general, that seems a significant limitation.

Correctness: The methods see correct.

Clarity: The manuscript is well-written. A key contribution of the manuscript is a collection of computational tricks that implements the proposed encoder. That is described in Section 3.1, and that would benefit by extending the description.

Relation to Prior Work: Relation to previous work is sufficiently clearly discussed.

Reproducibility: Yes

Additional Feedback: Eq. (1): Element-wise notation for state variable z_i is slightly misleading as z_i^T cannot be solved without solving all z_i simultaneously. R97: learns -> "defines"(?) ========================= UPDATE: I have read authors' response letter. Overall this seems a valid study that provides a solution to infer neural network parameterized ODE models from irregular and partially sampled (not-aligned) data. As far as I can tell, this method provides a solution to a problem that no other (single) method can handle. I will increase my overall score to 6.


Review 4

Summary and Contributions: This paper aims to learn dynamics of the continuous multi-agent system relying on the ODE based on the general framework of Latent ODE. The main contribution of this paper lies in the proposed reasonable way to extend the latent ODE from single-agent system to multi-agent system, in which the initial latent states for all objects are inferred jointly to take into account the coupling and relations between objects by the proposed temporal-aware GNN.

Strengths: 1. The proposed algorithm relying on temporal GNN extending the latent ODE from single-agent system to multi-agent system is reasonable. 2. The provided experiments are convincing.

Weaknesses: 1. The novelty from latent ODE to latent graph ODE is incremental. 2. More experiments about ablation study is preferable, e.g., 1) experiments on the comparison of with / without the attention mechanism of "dynamic node representation learning" in Equation 4, 2) with / without the positional encoding in "Message" in Equation 5. 3. More details should be provided for the ODE function g_i in Equation 8.

Correctness: Yes.

Clarity: The writing is moderately clear and good.

Relation to Prior Work: Yes.

Reproducibility: Yes

Additional Feedback: 1. More experiments about ablation study is preferable, e.g., 1) experiments on the comparison of with / without the attention mechanism of "dynamic node representation learning" in Equation 4, 2) with / without the positional encoding in "Message" in Equation 5. 2. More details should be provided for the ODE function g_i in Equation 8.


Review 5

Summary and Contributions: The authors develop a variational model combining a GNN-based encoder with latent variables satisfying an ODE. Here, the GNN-based encoder, which is inspired by the Transformer model, allows for partial observations and missing data, while the ODE models continuous system dynamics and allows for irregular observations in time. Experiments are conducted on three datasets: two synthetic, and one real involving motion capture data. The model performs well in MSE for interpolation and extrapolation in all experiments.

Strengths: The proposed model is general and sophisticated, but remains sufficiently tractable so as to be practical. Experiments are detailed and the improvements in MSE are impressive.

Weaknesses: There is too little discussion of related work here. For example, the authors “propose to learn system dynamics from irregularly-sampled and partially observed system [sic] for the first time”. This is quite a bold claim, considering the numerous previous efforts to accomplish exactly this task. See point 6 for more details. In particular, going beyond machine learning narrowly defined to statistics more generally defined, the paper disregards an enormous amount of previous statistical work. The presentation could also use improvement, with numerous grammatical issues throughout. Model choices could also use some additional explanation, especially in light of related work.

Correctness: Both the model and training procedure appear to be valid and sufficiently rich to tackle the problem discussed. Experiments are conducted appropriately.

Clarity: The paper is not especially well written. Another editing pass would likely serve to clean up a number of issues throughout.

Relation to Prior Work: Due to the bold claims regarding novelty, the authors open themselves up to comparisons with numerous other methods. There is a plethora of relevant literature that has not been addressed here. Some of the most recent of these include any of the existing methods of data imputation ([1] as an example, but there are many more), or continuous-time hidden Markov models [2]. The model presented here is also highly similar to graph neural ordinary differential equations (GNODE) [3], which were developed for a similar purpose. Perhaps a more focused presentation would help to avoid requiring many of these comparisons, but some discussion of the work relative to the GNODE framework is probably a good idea. [1] Mei, H., Qin, G., Eisner, J. (2019). Imputing Missing Events in Continuous-Time Event Streams. [2] Liu, Y.-Y., Li, S., Li, F., Song, L., Rehg, J. M. (2015). Efficient Learning of Continuous-Time Hidden Markov Models for Disease Progression. [3] Poli, M., Massaroli, S., Park, J., Yamashita, A., Asama, H., Park, J. (2019). Graph Neural Ordinary Differential Equations.

Reproducibility: Yes

Additional Feedback: Some assorted comments: - There are many incorrect usage/lack of plurals here — check each appearance of the word `system’. - How does the encoder relate to a graph attention network? - Some of the text in Figure 1 is quite small. I know space is at a premium, but can some of these be enlarged? - I find the notation a little confusing. Matrices are bolded, but collections of vectors (which are essentially the same object) are not. Overall, I think the paper would benefit from a notation section which explains these choices and others clearly. - A minor point, but the broader impact should mention the lack of interpretability of the model presented, especially given that the authors are targeting scientific applications. Line 70: Consider rewording the first sentence — the state is dictated by a set of time-dependent variables… Also, T should be defined. Line 125: “doc-product” -> “dot-product” Line 138: Please define ODESolve. Also what does the approximation sign mean? Line 170: exp should be Romanized

[Author Response · NeurIPS 2020]

We would like to thank all reviewers for their valuable and helpful suggestions. We first respond to the common request
on ablation study of our proposed encoder as shown in Table 1. Due to space limit, we will report results on other
datasets, more details about model description and related work, and revise typos in our final version as suggested.

**Ablation Study.** Our proposed encoder contains two modules: dynamic node representation network followed by a
temporal self-attention. The goal of the first module is to simultaneously capture spatial-temporal dependency among
nodes. We achieve this by introducing temporal dependency to spatial-based GNN with learnable positional encoding
and attention mechanism. To test the efficiency of each component, we remove them separately (LG-ODE-no att,
LG-ODE-no PE) and find the performances drop. This suggests that distinguishing the importance of nodes w.r.t
time and incorporating temporal information via learnable positional encoding would benefit model performance.
Additionally, to test the performance of adding learnable parameters and nonlinearity in positional encoding, we
compare with manually-designed positional encoding [15] (LG-ODE-fixed PE) and find our method more flexible
which produces more efficient temporal encoding. Secondly, to test the efficiency of temporal self-attention, we adopt
different sequence aggregation methods (LG-first, LG-mean) and find our method performs the best. This suggests that
nodes at different timestamp would represent different semantic meanings towards the initial state of the whole system.

Table 1: Mean Square Error (MSE) $\times 10^{-2}$ of Ablation Study and Baselines on Spring Dataset.

| Model | | LG-ODE | LG-ODE -no att | LG-ODE -no PE | LG-ODE -fixed PE | LG-ODE -first | LG-ODE -mean | Latent -ODE | NRI+RNN -imputation |
|---|---|---|---|---|---|---|---|---|---|
| Interpolation | 40% | **0.3350** | 0.5145 | 0.4431 | 0.4285 | 1.3017 | 0.3896 | 0.5454 | 2.0743 |
| | 60% | **0.3170** | 0.4198 | 0.4278 | 0.4445 | 1.1918 | 0.3901 | 0.5036 | 1.9857 |
| | 80% | **0.2641** | 0.4510 | 0.3879 | 0.4083 | 1.0796 | 0.3268 | 0.4290 | 1.9573 |
| Extrapolation | 40% | **1.7839** | 2.3847 | 1.7943 | 1.7905 | 6.5742 | 2.2499 | 6.6023 | 3.8966 |
| | 60% | 1.8084 | 2.1216 | 1.8172 | **1.7634** | 6.3243 | 2.1165 | 4.2478 | 3.8749 |
| | 80% | **1.7139** | 1.9634 | 1.7332 | 1.7545 | 5.7788 | 2.2516 | 4.3192 | 3.5762 |

**Reviewer 1.**

**A1. Interaction aspect of the model and Fig.2 explanation**. To show the importance of graph interaction, we
compare with Latent-ODE which processes each timeseries individually. Our model outperforms it over two tasks as
shown in Table 1. For Fig.2, we would like to clarify the terminology "interpolation" following the existing work [7].
We try to fit a curve using observed time points with a goal to minimize the MSE. Fig.2 plots the predicted curve as
interpolation results and these plotted prediction values may differ from the truth values that are conditioned on.

**A2. Experiments and limitations.** Thanks for your advice on experiments! We will add additional mocap sequences
and provide video link later as rebuttal allows no links. The limitation of our model is that we assume the graph structure
is fixed, but in reality graph structure also changes w.r.t time. We will leave it as a future work to further explore.

**Reviewer 2.** To the best of our knowledge, we are the first to handle irregularly-sampled partial observations with
known graph structure. The two papers you mentioned do not consider graph structure. The second paper only handles
irregularly-sampled data but not partially-observed dynamic system. It assumes all agents' observations are aligned.

**Reviewer 3.**

**A1.** To make our model comparable with existing ones, we compare with baselines from two problem variants. Firstly,
we employ RNN-imputation [R1] where the graph structure is not considered. It jointly imputes missing values
(interpolation) for all agents by simple concatenation of feature vectors. As shown in Table 1, the performance drops
which shows that such graph structure is essential for predicting interacting systems. Secondly, to show the effectiveness
of our way to handle irregularly-sampled partial observations, we combine RNN-imputation with NRI [2] where we
first impute each timeseries into regular-sampled one to make it a valid input for NRI, and then predict trajectories
jointly with graph structure (extrapolation). As shown in Table 1, the prediction error is large and one possible reason is
that we use estimated imputation values for missing data which would cause noise to NRI. Also the two-step process
separates imputation with prediction, whereas our approach is an end-to-end framework for both two tasks.

**A2.** For Eqn2, we adopt the GNN model in [2] to capture the interaction among agents. It firstly employs a shared
relation function $f_R$ to compute pair-wise influence, then employ a shared object function $f_O$ for influence aggregation.
Such weights sharing mechanism is commonly utilized in various GNN models [2,3,20].

**Reviewer 4.** We respectfully disagree with your comment that our model is incremental by extending Latent ODE. In
multi-agent system, despite each timeseries can be irregularly-sampled, such system can be only partially observed
(timeseries are not aligned). Also as agents continuously influence each other, how to combine such interaction with
irregular partial observations to make predictions remains challenging. Latent ODE only deals with single timeseries
and is not able to solve these problems. We therefore design a novel encoder that extracts spatial-temporal pattern from
irregularly-sampled partial observations and graph structure, and use it to infer all initial states simultaneously. The
whole system is then driven by a GNN that models continuous interaction among agents along time.

[R1] *Che, Z. et al. "Recurrent Neural Networks for Multivariate Time Series with Missing Values." Scientific reports vol. 8,1 6085. 2018*


[Meta-Review · NeurIPS 2020]

This paper proposes an approach to learning system dynamics when the time between observations is non-uniform and when the observations from different objects are not temporally aligned. There were concerns that a large body of related work was ignored.